# The Complete Mitochondrial Genomes of Two Rock Scallops (Bivalvia: Spondylidae) Indicate Extensive Gene Rearrangements and Adaptive Evolution Compared with Pectinidae

**DOI:** 10.3390/ijms241813844

**Published:** 2023-09-08

**Authors:** Fengping Li, Yu Zhang, Tao Zhong, Xin Heng, Tiancheng Ao, Zhifeng Gu, Aimin Wang, Chunsheng Liu, Yi Yang

**Affiliations:** 1School of Marine Biology and Aquaculture, Hainan University, Haikou 570228, China; lifengping_hnedu@163.com (F.L.); hnugu@163.com (Z.G.); aimwang@163.com (A.W.); lcs5113@163.com (C.L.); 2Sanya Nanfan Research Institute, Hainan University, Sanya 572025, China; 3Sanya Oceanographic Institution, Ocean University of China, Sanya 572000, China

**Keywords:** spondylid, phylogeny, adaptive evolution, positive selection, mitochondrial genome

## Abstract

Different from the diverse family Pectinidae, the Spondylidae is a small group with a single genus that shares the sedentary life habit of cementing themselves to the substrate. However, little information related to the genetic diversity of Spondylidae has been reported. In the present study, the complete mitochondrial genomes of *Spondylus versicolor* and *S. spinosus* were sequenced and compared with those of pectinids. The mtDNA of *S. versicolor* and *S. spinosus* show similar patterns with respect to genome size, AT content, AT skew, GC skew, and codon usage, and their mitogenomic sizes are longer than most pectinid species. The mtDNA of *S. spinosus* is 27,566 bp in length, encoding 13 protein-coding genes, 22 transfer RNA genes, and 2 ribosomal RNA genes, while an additional tRNA-*Met* was found in the mtDNA of *S. versicolor*, which is 28,600 bp in length. The monophylies of Spondylidae and Pectinidae were well supported, but the internal relationships within Pectinidae remain unresolved due to the paraphyly of the genus *Mimachlamy* and the controversial position of the tribe Aequipectinini. The gene orders of *S. versicolor* and *S. spinosus* are almost identical but differ greatly from species of the Pectinidae, indicating extensive gene rearrangements compared with Pectinidae. Positive selection analysis revealed evidence of adaptive evolution in the branch of Spondylidae. The present study could provide important information with which to understand the evolutionary progress of the diverse and economically significant marine bivalve Pectinoidea.

## 1. Introduction

Known as spiny or thorny oysters, Spondylidae are a family of marine bivalve mollusks that are most diverse in the tropical and subtropical areas of the Indo–West Pacific Ocean [1]. Spondylids are generally characterized by their spinose shells and strong hinge dentition, and they usually attach to the substratum with their left valve [2]. As the single genus within Spondylidae, *Spondylus* is composed of approximately 70 living species [3], some of which constitute an important fishing resource for local commercial consumption [4].

Along with family Pectinidae, Spondylidae were assigned to the superfamily Pectinoidea. The affinity between Pectinidae and Spondylidae has been supported by anatomical and morphological evidence. Both families share the characteristics of complex eyes around the mantle and a relatively well-developed nervous system [5]. The sperm ultrastructure has been considered as a useful tool for bivalve taxonomy and phylogenetic reconstruction [6], and a comparative sperm ultrastructure study indicated that spondylid spermatozoa strongly resembled those of scallops [1]. Paleontological data of shell morphology also indicated that spondylids originated from an ancestor within the Pectinidae [7].

The complete mitochondrial (mt) genome includes both sequence information and structural character, and it has been widely used in species delimitation and phylogenetic analyses [8,9,10,11] based on a serious of advantages such as high abundance, a lack of recombination, a higher rate of evolution, and maternal inheritance [12]. With more mitochondrial genomes sequenced, several traditionally recognized characters of animal mitogenomes have been rejected in terms of the discovery, for example, of doubly uniparental inheritance and homologous recombination [13,14,15]. Recent studies have also revealed the adaptive evolution of the mitochondrial protein-coding genes, which are important to oxygen usage and energy metabolism [16]. Some examples of adaptive evolution have been reported in the *atp8*, *nad2*, and *nad5* genes of the deep-sea Brisingida [17], the *nad4* gene of the freshwater species *Limnoperna fortune* [18], and the *atp6*, *cox1*, *cox3*, *cytb*, and *nad* 1–5 genes of the deep-sea alvinocaridid [19]. All spondylids share a cementing life habit in contrast to the diverse habit types which derived from the byssal attaching life habit within Pectinidae [20]. Currently, there are no studies exploring the adaptive evolution of mitochondrial genomes in pectinoids with different lifestyles that may be attributed to different mechanisms of oxygen consumption and energy metabolism.

Compared with other metazoans, mollusks are more prone to gene rearrangements [21,22,23]. In pectinids, mitochondrial gene rearrangements involve protein-coding genes (PCGs), ribosomal RNA (rRNA) genes, and transfer RNA (tRNA) genes, and considerable gene order changes could be detected even at the level of tribes [24]. However, the mitochondrial gene rearrangements within Pectinoidea other than Pectinidae have never been detected before.

In the present study, the complete mitogenomes of *S. versicolor* (Schreibers, 1793) and *S. spinosus* (Schreibers, 1793) were sequenced and compared with those of other pectinoids available on GenBank (Table 1). Our aims were as follows: (1) to characterize the mitogenomic features of Spondylidae; (2) to explore their gene orders; (3) to reconstruct a robust phylogenetic framework of Pectinoidea; and (4) to test whether adaptive evolution has occurred within Pectinoidea mitogenomes.

## 2. Results and Discussion

### 2.1. Mitogenomic Structure and Composition

The coverage plots of both species were generated by remapping all reads back to the assembled mitogenomes (Appendix A). An obvious coverage bump, which may be caused by repetitive sequences, was discovered in each mitogenome. However, the bump in *S. spinosus* mitogenome also fell in the region generated via Sanger sequencing (Appendix A), which supported the absence of repetitive sequences. The coverage bumps could be offset by adding different numbers of repeats according to the coverage of the reads; however, such a solution could lead to an excessive genome size that falls outside the normal range of Pectinoidea. Since the whole process of NGS assembly was visible and no errors were detected in the mapped reads, it is assumed that the bumps (with a relatively higher AT content values) in the non-coding regions of two mitogenomes might result from a flaw in NGS sequencing whereby regions with higher AT content values usually acquire more sequencing depth.

The mitochondrial genome composition and structure of the two newly sequenced spondylids are shown in Appendix A and Figure 1. The mitogenome lengths of *S. versicolor* and *S. spinosus* are 28,600 bp and 27,566 bp, respectively, which is longer than most pectinid species (ranging from 16,079 to 20,964 bp), except for that of *Placopecten magellanicus*, which is 32,115 bp in length (Table 1). The differences in mitogenome size are mainly caused by the variation of length of non-coding regions. The *S. spinosus* mtDNA encodes for 13 PCGs, 22 tRNA genes, and 2 rRNA genes, identical to those found in the typical metazoan mtDNA, while an additional tRNA-*Met* was found in the mtDNA of *S. versicolor*. Like all marine bivalves, all genes of *S. spinosus* and *S. versicolor* are encoded on the major strand [14]. The AT content values of *S. versicolor* and *S. spinosus* are 57.9% and 58.3%, respectively (Appendix A), indicating a high A + T bias. The AT skew values of *S. versicolor* and *S. spinosus* are −0.23 and −0.21, respectively, in contrast to the positive GC skew values 0.39 and 0.34, respectively (Appendix A). A similar tendency of asymmetry has also been revealed in different mollusk groups [25].

### 2.2. Protein-Coding Genes, Transfer RNA, and Ribosomal RNA Genes

Both *S. versicolor* and *S. spinosus* contain 13 PCGs, including the *Atp8* gene which is sometimes missing in bivalve mtDNA. The AT content values of PCGs of *S. versicolor* and *S. spinosus* range from 56.3% to 63.2% and from 50.5% to 66.7%, respectively (Appendix A). The negative AT skew and positive GC skew values of the PCGs and three individual positions show the same trend of asymmetry as the whole mitochondrial genome. Most PCGs of *S. versicolor* and *S. spinosus* start with the conventional initiation codons ATA and ATG, while the *Cox3*, *Atp8*, *Cytb*, and *Nad6* of *S. versicolor* and the *Cox3*, *Cytb*, *Nad4L*, *Nad4*, and *Atp6* of *S. spinosus* employ alternative start codons, namely, GTG, TTG, and ATT (Appendix A). All PCGs of *S. versicolor* and *S. spinosus* end with complete stop codons TAA and TAG, except for the *Cox2* gene, which uses the incomplete stop codon TA in both mitogenomes. The truncated stop codons (TA and T) might be completed by the TAA termini via post-transcriptional polyadenylation [26]. The mtDNA of *S. versicolor* contains 3711 codons (excluding all stop codons), compared with the 3694 codons of *S. spinosus* (Appendix A). Among all the codons in both mitogenomes, UUU (Phe) and GUU (Val) are the most frequently used. A synonymous codon usage bias was also discovered (Appendix A). Most of these preferred codons contain G and U, which could explain the positive GC skew and negative AT skew of the PCGs and whole mitogenome. Mutation pressure is the major force in determining codon usage patterns in mitochondrial genes [27].

Typical metazoan mtDNA are characteristic of 22 tRNA genes. However, the bivalve mtDNA usually show differences with respect to the number of tRNA genes. In this study, one additional tRNA-*Met* was only duplicated in the mtDNA of *S. versicolor*. The duplication of tRNA-*Met* has also been detected in the family Pectinidae [28,29], as well as in other bivalve taxa such as Ostreidae [30], Gryphaeidae [22], and Veneridae [31]. The length of all tRNA genes ranges from 62 to 75 bp (Appendix A). The AT content values of *S. versicolor* and *S. spinosus* are 46.5% and 48.7% (Appendix A), respectively. The negative AT skew values (−0.17 and −1.50) and positive GC skew values (0.23 and 0.24) show the same tendency of asymmetry as PCGs. All tRNA genes could be folded into clover-leaf secondary structures except for the tRNA-*Ser* (recognizing AGN) in both mitogenomes since the dihydrouracil (DHU) arm is missing (Appendix A). The absence of DHU arm is also typical of metazoan mtDNA [32].

The *rrnL* genes of *S. versicolor* and *S. spinosus*, located between tRNA-*Gln* and tRNA-*Phe,* are 1491 and 1460 bp in length, with AT content values of 54.9% and 54.5%, respectively. In contrast, *rrnS* genes are 935 and 928 bp, with AT content values of 53.7% and 54.0%, respectively. The *rrnS* genes are located between tRNA-*Phe* and tRNA-*Ala.* All rRNA genes show a positive AT skew (Appendix A).

### 2.3. Phylogenetic Analysis

The phylogenetic relationships of Pectinoidea were reconstructed using probabilistic methods (Figure 2). According to the Bayesian information criterion (BIC), the best partition scheme for PCGs was the one combining genes by subunits but analyzing each codon position separately, while the best partition scheme for rRNAs was the one combining the two genes (Appendix A). The evolutionary model for each scheme was shown in Appendix A. Both ML (−lnL = 145,621.21) and BI (−lnL = 141,451.52 for run 1; −lnL = 141,453.56 for run 2) analyses arrived at identical topologies (Figure 2).

In the present study, only two pectinoid families were included in the phylogenetic analyses, and their close affinity was undoubtedly revealed (Figure 2). However, the phylogenetic position of the Spondylidae to the other families in the Pectinoidea has been highly contentious. Morphological and paleontological data proposed two families, Pectinidae and Entoliidae, singly or in combination as sisters to the Spondylidae [7,32,33]. Similar to the result derived from morphological data, a COI-based phylogeny also supported the Pectinidae + Spondylidae relationship [34]. A recent study based on five nuclear and mitochondrial genes indicated the non-monophyly of the family Propeamussiidae that was divided into four lineages, with two subsets of species as the sister group to the Pectinidae and Spondylidae, respectively [35]. Since the previous molecular phylogenies were based on short gene fragments, some deep nodes were not well resolved. Therefore, it is necessary to include the mitogenomic data of a more comprehensive taxonomic sampling of the Propeamussiidae in order to resolve the phylogenetic relationships of the Pectinoidea.

Within the Pectinidae, most internal nodes were well supported (Figure 2). A total of 14 pectinid species (Table 1) formed three subfamilies, namely, Pedinae, Pectininae, and Palliolinae. Although the Pedinae is recovered as monophyletic, the monophyly of *Mimachlamys* within this subfamily is not supported due to the discrete positions of *M. varia* and *M. nobilis* + *M. senatoria*. The paraphyly of *Mimachlamy* has also been reported in previous phylogenies [20,24,36]. Another result worth mentioning is the phylogenetic position of the tribe Aequipectinini (represented by the genus *Argopecten*), which is placed within Pectininae (Figure 2). Even though the current topology is highly supported, it is inconsistent with a previous phylogeny in which Aequipectinini was placed in the basal clade within Pectinidae [36]. Malkócs et al. [24] concluded that different molecular markers could result in different topologies of Pectinidae, as indicated in other mollusk groups [25,37]. When the mitochondrial PCGs were used for phylogenetic analysis, Aequipectini always formed within Pectininae and in accordance with our result, while the topology derived from mitochondrial rRNA and nuclear H3 genes supported that of Puslednik and Serb [36]. The above contradiction in pectinid topologies was explained via the limited taxon sampling and different choice of outgroups [24,36]. The family Pectinidae contains around 350 living species [24], only 14 of which are provided with mitogenomes. Arriving at a well-resolved phylogenetic relationship within Pectinidae still relies on comprehensive taxon sampling.

### 2.4. Gene Rearrangement

The PCG order of the mollusk mitogenome is generally more conserved in closely related species compared with tRNAs, of which duplication and rearrangement are quite common [38]. Thus, only the PCG order is kept to determine the possible scenarios of rearrangement, as previous studies have suggested [39]. The two species, *S. versicolor* and *S. spinosus*, share an identical PCG order (Figure 3), suggesting a potential ancestral characteristic of the Spondylidae. However, little information is known with respect to the internal relationships of Spondylidae, and more mitogenomic data are needed in order to verify this hypothesis in the future.

Different from the Spondylidae, the family Pectinidae possesses a rich species diversity and shows considerable PCG rearrangements. A total of seven different PCG orders are observed within Pectinidae (Figure 3). The subfamily Pectinidae consists of three lineages which correspond to three unique gene orders, respectively. The first lineage grouped by *Pecten* and *Amusium* shares the same gene order, which is almost identical to that of the second lineage represented by the genus *Argopecten*, as only one gene translocation is found between the two sister groups. The third lineage contains a single species, *Ylistrum balloti*, which is placed at the basal position of Pectininae. However, the PCG order of *Y. balloti* shows only one gene block (*Atp6*–*Nad1*–*Cox1*) shared with those of the other two lineages. As the sister group of Pectininae, the subfamily Palliolinae is represented by a single species: *Placopecten magellanicus*. Limited taxon sampling restricts our understanding of PCG order rearrangements as no shared gene block is found between Palliolinae and Pectininae.

The subfamily Pedinae contains three types of PCG orders, of which the differences only lie in the location of *Nad5* gene (Figure 3). One of the two major evolutionary clades grouped by *Mimachlamys nobilis* and *M. senatoria* shares an identical PCG order, which is also possessed by *Mizuhopecten yessoensis*, which is placed in the other clade. Variation among the second clade can best be explained by assuming the order in *M. yessoensis* as pleisomorphic, with one translocation of *Nad5* leading to *Crassadoma gigantea* + *Chlamys farreri* and another translocation leading to *M. varia*.

### 2.5. Positive Selection Analysis

In the analysis of branch-specific models for the 12 concatenated PCGs, the ω (Ka/Ks) ratio under the M0 (“one-ratio”) model is 0.03351, indicating strong functional constraints. However, the ω ratio averaged over all lineages is almost never > 1 since positive selection is unlikely to affect all sites over a prolonged period of time. The M1 (“free-ratios”) model fits the data significantly better than the M0 model (*p* < 0.01, Table 2), suggesting that the mitochondrial PCGs have been subject to different selection pressures in different pectinoid lineages. In addition, the M2 (“two-ratios”) model was found to be significantly better than the M0 model (*p* < 0.01, Table 2) when setting the family Spondylidae as a foreground branch, and the ω ratio (ω1 = 1.84124) supports positive selections on the Spondylidae branch.

Although animal mitogenomes have long been understood to undergo purifying selection, it is possible that they can also evolve under weak and/or episodic positive selection in response to shifts in physiological or environmental demands [40,41]. For example, the episodes of positive selection detected in the terrestrial panpulmonate mitogenomes suggested adaptations from marine to non-marine habitats [42]. A similar result was also achieved in the mitogenome of freshwater mussels [18]. Another study focusing on the genus *Acrossocheilus*, a fish group that inhabits from the tropical to subtropical regions, detected positive selection in the mitogenomes of this group and suggested evolutionary adaptations to different temperatures [43]. Furthermore, episodic positive selections have been widely reported in the mitogenomes of deep-sea organisms, suggesting adaptations to extreme environments [17,19,44].

The positive selection was found on the branch of Spondylidae against Pectinidae, indicating plenty of variations between the protein-coding genes of the two families. The Spondylidae and the Pectinidae possess quite different life habits. All members of the Spondylidae cement to substrates, while byssal attachment is the major habit for the Pectinidae [20]. The different life habits are related to different ecological requirements and behavioral attributes [45], which could lead to different mechanisms of energy metabolism. Therefore, the mitochondrial protein-coding genes which are related to the oxygen usage and energy metabolism of animals might have evolved in different directions. On the other hand, the sister relationship between the Spondylidae and the Pectinidae has been challenged in previous phylogenies [35], and their phylogenetic positions within Pectinoidea may change with the inclusion of more families. Since CodeML analysis is sensitive to tree topology, future studies with broader taxon sampling within Pectinoidea are needed in order to verify the present observed positive selection on the branch of Spondylidae within Pectinoidea.

## 3. Materials and Methods

### 3.1. Sample Collection and DNA Extraction

The specimens of *S. versicolor* and *S. spinosus* were collected on Ximaozhou Island (18°14′22″ N; 109°22′42″ E) and Wuzhizhou Island (18°18′55″ N; 109°46′3″ E), respectively. The soft tissue of the specimen was preserved in 95% ethanol in the Laboratory of Economic Shellfish Genetic Breeding and Culture Technology (LESGBCT), Hainan University.

Total genomic DNA was extracted from ethanol-fixed tissue of adductor muscle using TIANamp Marine Animals DNA Kit (Tiangen, Beijing, China), following the manufacturer’s the instructions, and stored at −20 °C.

### 3.2. DNA Sequencing, Mitogenome Assembly and Annotation

Genomic DNA of *S. versicolor* and *S. spinosus* was sent to the Novogene Company (Beijing, China) for library construction and high-throughput sequencing. The DNA library was generated using NEB Next^®^ Ultra™ DNA Library Prep Kit for Illumina (NEB, Ipswich, MA, USA) according to the manufacturer’s instructions. A total of two libraries with insert sizes of approximately 300 bp were prepared and then sequenced as 150 bp paired-end runs on the Illumina NovaSeq 6000 platform. Finally, 26,328,249 clean reads of each direction were generated for the library of *S. versicolor*, and 18,244,235 clean reads were generated for *S. spinosus*. The generated clean data were imported in Geneious Prime 2021.0.1 [46] for mitogenome assembly, with the strategy following Irwin et al. [37].

In order to eliminate the influence of the unambiguous regions derived from the next-generation data, we designed two pairs of primers and employed long-PCR amplifications to fill the assembled gap using the SveF forward (5′-GTGGGGGTTGGCTTAAAGTGGATTTAGG-3′) and SveR reverse (5′-CAAACTCCCACTAAAGACAGGCCTAGCA-3′) specific primers for *S. versicolor*, and SspF forward (5′-GGTCTCCATAAAGTGGGACTTGTCAGTG-3′) and SspR reverse (5′-CCACACCCCACTATCACCAAATCGCTT-3′) specific primers for *S. spinosus*. The location of the primers was indicated in Appendix A.

The PCR reactions contained 2.5 μL of 10× buffer (Mg^2+^ plus), 0.5 μL of dNTPs (10 mM), 1 μL of each primer (10 μM), 0.8 μL of template DNA (25–40 ng/μL), 0.25 μL of TaKaRa Taq DNA polymerase (1U), and DEPC water up to 25 μL. PCR reactions were conducted by an initial denaturation step at 94 °C for 3 min, followed by 35 cycles of 30 s at 94 °C, 30 s at 50 °C, and 2 min at 72 °C; then, a final extension step at 72 °C for 10 min was undertaken. The PCR products were purified via ethanol precipitation and sequenced at Beijing Liuhe BGI (Beijing, China). The length of the two fragments was 1581 bp for *S. versicolor* and 415 bp for *S. spinosus*, and only the forward and reverse PCR primers were used as Sanger sequencing primers from both directions. The Sanger sequencing data of each species were assembled using SeqMan (www.DNASTAR.com, accessed on 1 July 2023). The amplified fragments were then imported into Geneious Primer to reconstruct complete and circular molecules. The short reads were remapped back to the assembled mitogenomes by Geneious Prime using the following parameters: custom sensitivity with a minimum mapping quality of 95%, a maximum mismatch of 5%, and fine-tuning 3 times.

The PCGs and rRNA genes were determined via ORF Finder (http://www.ncbi.nlm.nih.gov/orffinder, accessed on 1 July 2023), MITOS Webserver [47], and BLAST (http://blast.ncbi.nlm.nih.gov/Blast.cgi, accessed on 1 July 2023). The secondary structure of tRNA genes was predicted by MITOS and ARWEN [48]. The annotated complete mitogenomes were submitted to GenBank with the accession numbers OR167109 for *S. versicolor* and OR167110 for *S. spinosus*. The mitochondrial gene map was generated with CGView [49].

### 3.3. Phylogenetic Analysis

A total of 16 pectinoids were included in phylogenetic analyses (Table 1), with 2 oysters as outgroup. The species from GenBank were identified by referring to the original publication. The dataset contained the nucleotide sequences of the 12 PCGs (*Atp8* was not included) and 2 rRNA genes. The PCGs were aligned separately as codons using ClustalW integrated in MEGA X [50]. The rRNA genes were aligned separately with MAFFT v.7 [51]. The ambiguously aligned positions of rRNA genes were removed using Gblocks v.0.91b [52] with default parameters. All separated alignments were finally concatenated into a single dataset in Geneious Prime and then converted into different formats using DAMBE5 [53] for further analyses.

Phylogenetic analyses were conducted using maximum likelihood (ML) and Bayesian inference (BI) analyses. ML analyses were performed via IQtree 1.6.10 [54], allowing partitions to have different evolutionary rates (-spp option) with 10,000 ultrafast bootstrap pseudoreplications (-bb option). BI were conducted using MrBayes 3.2.6 [55], running four simultaneous Monte Carlo Markov chains (MCMC) for 10,000,000 generations, sampling every 1000 generations, and discarding the first 25% generations as burn-in. Two independent BI runs were carried out to increase the chance of adequate mixing of the Markov chains and to increase the chance of detecting failure to converge. The effective sample size (ESS) values of all parameters calculated by Tracer v1.6 were above 200.

The best partition schemes and best-fit substitution models for BI analyses were identified using PartitionFinder 2 [56] under the BIC. For PCGs, the partitions tested were all genes combined, all genes separated (except *nad4-nad4L*), and genes grouped by subunits (*Atp*, *Cytb*, *Cox*, and *Nad*). Additionally, these three partition schemes were tested, considering the three codon positions separately. The rRNA genes were analyzed with two different schemes (genes grouped or separated). The best partition schemes for ML analyses were also determined via PartitionFinder 2, while the best-fit substitution models were calculated with ModelFinder [57] as implemented in IQ-TREE 1.6.12. The best-fit substitution schemes and models are provided in Appendix A.

### 3.4. Gene order Comparison and Positive Selection Analysis

The mtDNA of *S. versicolor* and *S. spinosus*, together with those of other pectinids published before (Table 1). were used for PCG rearrangement analysis. The PCG orders of each species were conducted and visualized using Microsoft Visio 2016. The *Atp8* gene was excluded since it was missing in several previously published mitogenomes.

To evaluate the potential adaptive evolution in the PCGs of spondylids, the branch model tests were performed to analyze the difference of selective pressure between the cementing Spondylidae and Pectinidae in the CODEML program of the PAML package [58]. The “one-ratio” model (M0), “free-ratio” model (M1), and “two-ratio” model (M2) were used in the combined dataset of 12 PCGs.

## 4. Conclusions

The mtDNA of *S. versicolor* and *S. spinosus* show similar patterns with respect to genome size, AT content, AT skew, GC skew, and codon usage, and their mitogenomic sizes are longer than those of most pectinid species. The gene orders of *S. versicolor* and *S. spinosus* are almost identical but differ greatly from species of the Pectinidae, indicating extensive gene rearrangements compared with Pectinidae. The internal phylogenetic relationships within Pectinoidea remain unresolved and call for broader taxon sampling. Evidence of adaptive evolution in the branch of Spondylidae was also detected. The newly sequenced molecules could provide important information with which to understand the evolutionary progress of the diverse and economically significant marine bivalve Pectinoidea.

## Figures and Tables

**Figure 1 ijms-24-13844-f001:**
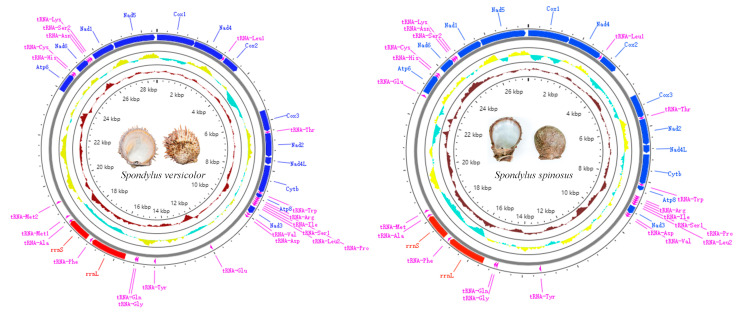
Mitochondrial genome maps of *S. versicolor* and *S.spinosus*. The innermost circle indicates GC skew values, while the adjacent outer circle represents GC content.

**Figure 2 ijms-24-13844-f002:**
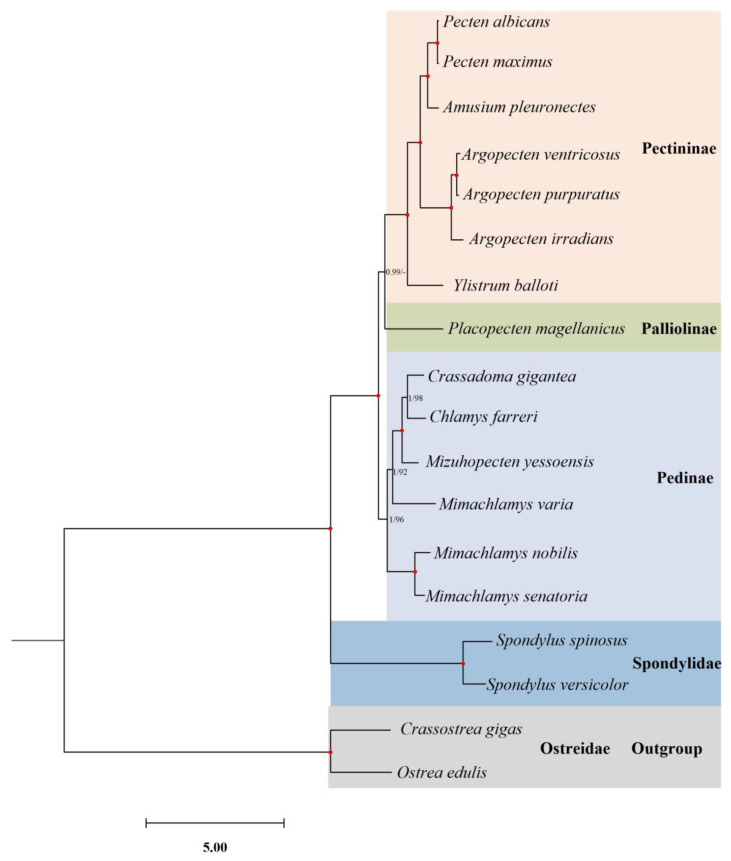
Phylogenetic relationships of Pectinoidea based on the concatenated nucleotide sequences of 12 mitochondrial protein-coding genes. The reconstructed Bayesian inference (BI) phylogram using *Crassostrea gigas* and *Ostrea edulis* as the outgroup is shown. The first number at each node is the Bayesian posterior probability (PP), and the second number is the bootstrap proportion (BP) of maximum likelihood (ML) analyses. Nodes with maximum statistical supports (PP = 1; BP = 100) are marked with a solid red circle. BP values under 90 are marked with a dash line.

**Figure 3 ijms-24-13844-f003:**
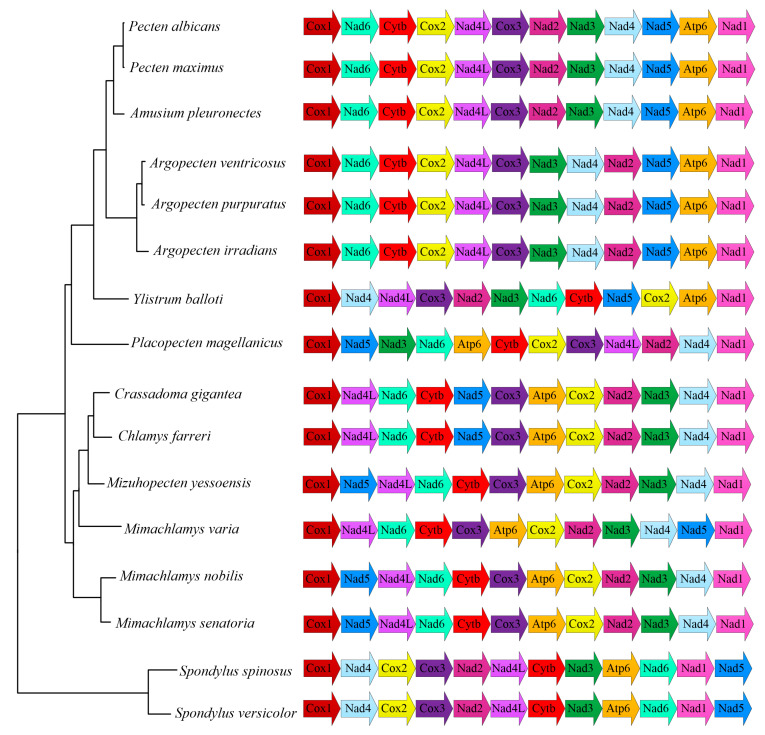
Linearized mitochondrial gene orders of the represented pectinoid mitogenomes available on GenBank (accession numbers are provided in Table 1).

**Table 1 ijms-24-13844-t001:** List of mtDNA used in the present study.

Newly Sequenced Mitochondrial Genome				
Family	Species	Length (bp)	Sampling Time	Accession No.
Spondylidae	*Spondylus versicolor*	28,600	May, 2022	OR167109
Spondylidae	*Spondylus spinosus*	27,566	May, 2022	OR167110
**GenBank Mitochondrial Genomes**				
**Family**	**Species**		**Length (bp)**	**Accession No.**
Pectinidae	*Pecten albicans* (Schröter, 1802)		16,653	KP900974
Pectinidae	*Pecten maximus* (Linnaeus, 1758)		17,252	KP900975
Pectinidae	*Amusium pleuronectes* (Linnaeus, 1758)		18,044	MT419374
Pectinidae	*Argopecten ventricosus* (G. B. Sowerby II, 1842)		16,079	KT161261
Pectinidae	*Argopecten purpuratus* (Lamarck, 1819)		16,270	KT161260
Pectinidae	*Argopecten irradians* (Lamarck, 1819)		16,212	KU589290
Pectinidae	*Ylistrum balloti* (Bernardi, 1861)		19,484	ON041136
Pectinidae	*Placopecten magellanicus* (Gmelin, 1791)		32,115	DQ088274
Pectinidae	*Crassadoma gigantea* (J. E. Gray, 1825)		18,495	MH016739
Pectinidae	*Chlamys farreri* (K. H. Jones and Preston, 1904)		20,889	EF473269
Pectinidae	*Mizuhopecten yessoensis* (Jay, 1857)		20,964	FJ595959
Pectinidae	*Mimachlamys varia* (Linnaeus, 1758)		20,400	MZ520326
Pectinidae	*Mimachlamys nobilis* (Reeve, 1852)		17,935	FJ595958
Pectinidae	*Mimachlamys senatoria* (Gmelin, 1791)		17,383	KF214684
Ostreidae	*Crassostrea gigas* (Thunberg, 1793)		18,225	EU672831
Ostreidae	*Ostrea edulis* (Linnaeus, 1758)		16,320	JF274008

**Table 2 ijms-24-13844-t002:** CODEML analyses of selection on mitochondrial genes in the Spondylidae lineage.

Branch-Specific Models				
Model	lnL	Estimates of parameters	Model compared	2ΔlnL
M0	−100,792.0493	ω = 0.03351		
Two-ratio	−100,751.3274	ω1 = 1.84124, ω0 = 0.03263	Two-ratio versus M0	81.4439 **
Free-ratio	−100,397.5439		Free-ratio versus M0	789.0108 **

** *p* < 0.01.

## Data Availability

The newly sequenced mitogenomes in the present study have been deposited in GenBank (OR167109-OR167110).

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
