# Peer review of "The Complete Mitochondrial Genomes of Two Rock Scallops (Bivalvia: Spondylidae) Indicate Extensive Gene Rearrangements and Adaptive Evolution Compared with Pectinidae"

_ijms, 2023, doi:10.3390/ijms241813844_

Round 1

Reviewer 1 Report

-Considering that only two specimens of two species of the speciose genus Spondylus were sequenced (and only their mitogenomes) and only a small part of the members of the highly diverse are covered by additional sequences taken from Genbank, I am afraid that the manuscript lacks impact and significance to be published in a journal like IJMS. Here is an excerpt from the journal’s website:

”Scope

In the International Journal of Molecular Sciences, molecules are the object of study; among those studies, we find:

  • Fundamental theoretical problems of broad interest in biology, chemistry and medicine;
  • Breakthrough experimental technical progress of broad interest in biology, chemistry and medicine;
  • Application of the theories and novel technologies to specific experimental studies and calculations.“

Therefore, I am afraid that the manuscript lacks significance and novelty to be considered for publication in the IJMS. Nevertheless, it is surely worth to be published in another journal, maybe a journal focusing more on the publication of papers on whole mitogenome sequencing of only one or two species.

In the following,  am providing some more detailed comments that might be helpful for improving the manuscript:

-The second part of the title gives the impression that a comprehensive analysis of the superfamily Pectinoidea was carried out. However, the only new data in the manuscript are related to two species of Spondylus. Although the manuscript still contains valuable data, the title should be rephrase to describe more clearly the new data provided in the manuscript. As all other sequences were taken from GenBank, this is not new data. Furthermore, only a tiny part of the highly diverse superfamily Pectinoidea is covered at all. Please be more critical with the content that the manuscript includes throughout and please do not give the impression that it contains more than that (also applicable to the Abstract and the rest of the manuscript).

-There are several minor mistakes throughout the manuscript. Please be sure to correct such mistakes, e.g. Table 1: “Spondylu”; Figure 1 and caption to Figure 1: “Spondylu”; Header of Table 2: “Spondylu”; Figure 2: “Spondlus”; caption to Figure 2: “Spondylu”; Header of Table 4: “Spondylu” etc.

- Please indicate how the correct identification of the samples from GenBank was ensured.

-Add species authorities for all species upon first mentioning.

-Genus names should be abbreviated with the first letter and not with the first three letters (lines 176–177).

-The author contribution statement is missing.

The language quality is mostly ok, but there are many minor mistakes throughout (please see the detailed comments).

Author Response

Dear Reviewer,

First, we would like to say thank you for reviewing our manuscript and for your valuable comments. Our responses to the comments are shown below.

Comment 1: Considering that only two specimens of two species of the speciose genus Spondylus were sequenced (and only their mitogenomes) and only a small part of the members of the highly diverse are covered by additional sequences taken from Genbank, I am afraid that the manuscript lacks impact and significance to be published in a journal like IJMS.

Response 1: It is known that IJMS is a high-level journal, and we understand the concern of the Reviewer. However, the section we submitted belongs to a special issue entitled "Mitochondrial Genome of Aquatic Animals: Analysis of Structure, Evolution and Diversity". More information about this special issue could be found according to the following website https://www.mdpi.com/journal/ijms/special_issues/P9J4DJGUON. We believe that our manuscript fits the field of this special issue and hope the Editor and Reviewer to consider its publication.

Comment 2: The second part of the title gives the impression that a comprehensive analysis of the superfamily Pectinoidea was carried out. However, the only new data in the manuscript are related to two species of Spondylus. Although the manuscript still contains valuable data, the title should be rephrase to describe more clearly the new data provided in the manuscript. As all other sequences were taken from GenBank, this is not new data. Furthermore, only a tiny part of the highly diverse superfamily Pectinoidea is covered at all. Please be more critical with the content that the manuscript includes throughout and please do not give the impression that it contains more than that (also applicable to the Abstract and the rest of the manuscript).

Response 2: We agree with this point. In order to be more critical, we revised the title as the following “The complete mitochondrial genomes of two rock scallops (Bivalvia: Spondylidae) indicate extensive gene rearrangements and adaptive evolution compared with Pectinidae”. Similar revisions have also been made in the Abstract and other part of the manuscript.

Comment 3: There are several minor mistakes throughout the manuscript. Please be sure to correct such mistakes, e.g. Table 1: “Spondylu”; Figure 1 and caption to Figure 1: “Spondylu”; Header of Table 2: “Spondylu”; Figure 2: “Spondlus”; caption to Figure 2: “Spondylu”; Header of Table 4: “Spondylu” etc.

Response 3: These mistakes have been revised throughout the manuscript.

Comment 4: Please indicate how the correct identification of the samples from GenBank was ensured.

Response 4: The species from GenBank were identified by referring to their original publication.

Comment 5: Add species authorities for all species upon first mentioning.

Response 5: The species authorities of the two Spondylus species have been added at the last paragraph of Introduction (where it was first mentioning), while those for the Pecinidae as well as outgroups were added in Table 1.

Comment 6: Genus names should be abbreviated with the first letter and not with the first three letters (lines 176–177).

Response 6: The first three letters for genus abbreviation has been revised as first letter.

Comment 7: The author contribution statement is missing.

Response 7: The author contribution statement is added at the end of this manuscript.

Reviewer 2 Report

Dr Li and Collaborators have determined the complete sequence of the mitochondrial genome in two species of Spondylidae. They further produced a phylogenetic tree to study the placement of the two new genomes in the context of Pectinoidea and an overview of different gene orders in the group. Main finding, according the the Authors, is the identification of positive selection acting on the mitochondrial genome of Spondylidae, that they relate to the cementing lifestyle of the group.

Positive selection:

The Authors make a strong point on the possibility that Spondylidae mitochondrial genomes may be evolving under positive selection. This hypothesis is based on a fairly minimalistic CODEML analysis. Furthermore, no support is to be found for their interpretation of a 'modified and adapted mechanism(s) of oxygen consumption and energy metabolism related to the unique cementing live habit of Spondylidae' (abstract).

Given the considerable interest that linking sequence level evolution of the mitochondrial genome with a specific functional adaptation may have, I would strongly encourage the Authors to improve this part to the point that it is fully convincing. If this is not in their plans/possibilities, I would substantially reduce their claims.

- Please consider moving the basic statistics of the two genomes (base composition, skews, codon usage) to supplementary material. The tables with gene positions grossly duplicate the information that is fund in GenBank files, and should be similarly transferred to supplementary materials. tRNA structures may go to supplementary materials as well (both species). Generally speaking, these information become interesting, and worth presenting in the main body of a paper and discussing in detail, only if they provide evidence for something else (unexpected biases, uncommon structures, ....), not as a series of archival tables.

-Some taxonomic names are misspelled, please revise.

-Line 56: please rephrase not to provide the misleading impression that citation 16 supports selection associated to energy metabolisms. This is a well known, and highly cited, review of gene order rearrangements in animals.

-Fig.1: please expand the caption to include an indication of the meaning of inner circles.

-Paragraph 2.4: this section describes the GO observed in Pectinidae as it emerges from the annotation of genomes already presented in the literature. As is, this cannot be regarded as an independent phylogenetic analysis that can support (or question) the phylogenetic tree based on sequence analysis. If the Authors wish to deploy GO in phylogenetic terms, a proper phylogenetic analysis (on GOs) may be conducted.

-Line 230-232: sharing of a plesiomorphic GO (or a GO hypothesized as plesiomorphic) cannot be regarded as an indication of monophily.

-Lines 253-255: I am not sure I understand the message correctly. Please rephrase.

-Line 259: the difference does not look as evident as to be taken as self-evident. Remember that the two nodes connect to a cluster of multiple subfamilies on one side and to a species poor genus on the other.

-Lines 288-300: I understand that an additional amplification/sequencing was performed to confirm a part of the genomic sequence that was deemed unreliable based on NGS assembly alone. Nevertheless the level of detail, and the absence of any reference to this point in the results section, does not allow to evaluate this aspect correctly. Please indicate a) which is the region under scrutiny and which is the indication that it may be problematic; b) which portion of the genome was amplified/sequenced and how was it sequenced (Sanger? primer walking?); c) how the new sequence compares to the primary assembly; c) if the primary assembly was not complete and circular from the beginning, which steps were taken to reconstruct a complete circular molecule from the assembly+amplified fragment.

-Line 301 (around): please provide (as supplementary material) a coverage plot obtained by remapping all reads over the two final sequences. In the case of evident changes/drops in coverage, please provide a justification.

-Lines 325-333: the strategy used to identify the best partitioning scheme is not totally clear. As the software proceeds by reading a list of (small, minimal) partitions provided by the user and then joins them in a smaller number of groups (if this does not significantly impair the fit of the model) the information to report is, in my view, a) the list of partitions submitted as starting point; b) the partitions as hypothesized by the software (i.e. how the starting partitions were joined); and c) the evolutionary model assigned.

-Table 6: please provide a key for the meaning of '**' or report p-values in the table.

Language is appropriate. Some minor stylistic corrections may be introduced at the copy-editing stage.

Author Response

Dear Reviewer,

First, we would like to say thank you for the comments provided for our manuscript. Those comments are all valuable and helpful for improving our manuscript. We have studied the comments carefully and made corrections in this manuscript. The responses are shown below.

Comment 1: Positive selection:

The Authors make a strong point on the possibility that Spondylidae mitochondrial genomes may be evolving under positive selection. This hypothesis is based on a fairly minimalistic CODEML analysis. Furthermore, no support is to be found for their interpretation of a 'modified and adapted mechanism(s) of oxygen consumption and energy metabolism related to the unique cementing live habit of Spondylidae' (abstract).

Given the considerable interest that linking sequence level evolution of the mitochondrial genome with a specific functional adaptation may have, I would strongly encourage the Authors to improve this part to the point that it is fully convincing. If this is not in their plans/possibilities, I would substantially reduce their claims.

Response 1:We agree with this point. However, we were not able to perform the functional analyses of mitochondrial PCGs. And therefore, as the Reviewer suggested, we have removed the claim that a 'modified and adapted mechanism(s) of oxygen consumption and energy metabolism related to the unique cementing live habit of Spondylidae' in the abstract.

Comment 2: Please consider moving the basic statistics of the two genomes (base composition, skews, codon usage) to supplementary material. The tables with gene positions grossly duplicate the information that is fund in GenBank files, and should be similarly transferred to supplementary materials. tRNA structures may go to supplementary materials as well (both species). Generally speaking, these information become interesting, and worth presenting in the main body of a paper and discussing in detail, only if they provide evidence for something else (unexpected biases, uncommon structures, ....), not as a series of archival tables.

Response 2: As the Reviewer recommended, we have removed the tables and figures mentioned above to supplementary materials.

Comment 3: Some taxonomic names are misspelled, please revise.

Response 3: We have checked through the manuscript and revised the misspelled taxonomic names.

Comment 4: Line 56: please rephrase not to provide the misleading impression that citation 16 supports selection associated to energy metabolisms. This is a well known, and highly cited, review of gene order rearrangements in animals.

Response 4: We have replaced the citation 16 to avoid the misleading impression. The information of new citation is as following:

[16] Da Fonseca, R.R.; Johnson, W.E.; O'Brien, S.J.; Ramos, M.J.; Antunes, A. The adaptive evolution of the mammalian mitochondrial genome. BMC Genomics 2008, 9, 119. https://doi.org/10.1186/1471-2164-9-119.

Comment 5: Fig.1: please expand the caption to include an indication of the meaning of inner circles.

Response 5: We have expanded the caption to explain the meaning of inner circles: “The innermost circle indicates GC skew values, while the adjacent outer circle represents GC content.”

Comment 6: Paragraph 2.4: this section describes the GO observed in Pectinidae as it emerges from the annotation of genomes already presented in the literature. As is, this cannot be regarded as an independent phylogenetic analysis that can support (or question) the phylogenetic tree based on sequence analysis. If the Authors wish to deploy GO in phylogenetic terms, a proper phylogenetic analysis (on GOs) may be conducted.

Response 6: We agree with this point. Following this advice, we have removed some views related to the gene order supporting or questioning the phylogenetic tree in the manuscript. However, we kept some descriptions of gene order characteristics of Pectinidae and Spondylidae in order to provide some information for future phylogenetic analysis on GOs.

Comment 7: sharing of a plesiomorphic GO (or a GO hypothesized as plesiomorphic) cannot be regarded as an indication of monophily.

Response 7: We have deleted this point.

Comment 8: Lines 253-255: I am not sure I understand the message correctly. Please rephrase.

Response 8: We intended to express that considerable evidence regarding positive selection in the mitogenome of Spondylidae compared with Pectinidae has been found in the present study. We carefully considered that this sentence is prone to ambiguity as the Reviewer mentioned, so we deleted it.

Comment 9: Line 259: the difference does not look as evident as to be taken as self-evident. Remember that the two nodes connect to a cluster of multiple subfamilies on one side and to a species poor genus on the other.

Response 9: We totally agree, and we have removed this point as the Reviewer suggested.

Comment 10: Lines 288-300: I understand that an additional amplification/sequencing was performed to confirm a part of the genomic sequence that was deemed unreliable based on NGS assembly alone. Nevertheless the level of detail, and the absence of any reference to this point in the results section, does not allow to evaluate this aspect correctly. Please indicate a) which is the region under scrutiny and which is the indication that it may be problematic; b) which portion of the genome was amplified/sequenced and how was it sequenced (Sanger? primer walking?); c) how the new sequence compares to the primary assembly; c) if the primary assembly was not complete and circular from the beginning, which steps were taken to reconstruct a complete circular molecule from the assembly+amplified fragment.

Response 10: This comment is valuable. The NGS data were assembled by Geneious Prime, which is visible during the process. For both mitogenomes, no more NGS data were able to be mapped to the template at certain length. We assume the reason may be caused by the influence of secondary structures, which could be indicated by the bump shown in coverage plot map (Figure S1).

The portion of the genome which was amplified was indicated in Figure S1 (by labelling the F and R primers in the mitogenome). The length of the two fragment was 1,581 bp for S. versicolor and 415 bp for S. spinosus, and only the forward and reverse PCR primers were used as Sanger sequencing primers from both directions.

The Sanger sequencing data of each species were assembled using SeqMan (www.DNASTAR.com). The amplified fragments were then imported into Geneious Primer to reconstruct complete and circular molecules by finding the location of primers as well as referring to the overlaps.

Comment 11: Line 301 (around): please provide (as supplementary material) a coverage plot obtained by remapping all reads over the two final sequences. In the case of evident changes/drops in coverage, please provide a justification.

Response 11: The coverage plot map was provided in the supplementary material as Figure S1. The coverage plots of both species were generated by remapping all reads back to the assembled mitogenomes (Figure S1). An obvious coverage bump was discovered in each mitogenome. This could be probably caused by the repetitive sequences in the mitochondrial genome. However, the bump in S. spinosus mitogenome fell in the region generated by Sanger sequencing (Figure S1), and therefore rejected the possibility that it was caused by repetitive sequences. Since the bumps of the two mitogenomes are almost in the same non-coding regions, they are assumed to be caused by certain secondary structure (e.g., the control region) that is difficult to be de novo assembled and may result in higher coverage by similar sequences in the nuclear genome.

Comment 12: Lines 325-333: the strategy used to identify the best partitioning scheme is not totally clear. As the software proceeds by reading a list of (small, minimal) partitions provided by the user and then joins them in a smaller number of groups (if this does not significantly impair the fit of the model) the information to report is, in my view, a) the list of partitions submitted as starting point; b) the partitions as hypothesized by the software (i.e. how the starting partitions were joined); and c) the evolutionary model assigned.

Response 12: The best partitioning scheme was determined by PartitionFinder 2. Since the software could also provide evolutionary model, and therefore we used the best partitioning schemes and the evolutionary models provide by PartitionFinder 2 for BI analysis. For ML analysis performed by IQ-TREE, we only employed the best partitioning schemes, while for the evolutionary models we selected the result calculated in ModelFinder implemented in IQ-TREE.

  1. a) the list of partitions was mentioned in the manuscript as “For PCGs, the partitions tested were all genes combined, all genes separated (except nad4-nad4L), and genes grouped by subunits (Atp, Cytb, Cox and Nad). Additionally, these three partition schemes were tested considering separately the three codon positions. The rRNA genes were analyzed with two different schemes (genes grouped or separated)”.
  2. b) the best partition was selected from the three schemes for PCGs and two schemes for rRNAs mentioned above, and the best partition scheme for PCGs was the one combining genes by subunits but analyzing each codon position separately, while the best partition scheme for rRNAs was the one combining the two gene.
  3. c) the evolutionary models for all schemes were provided in supplementary material Table S5.

Comment 13: Table 6: please provide a key for the meaning of '**' or report p-values in the table.

Response 13: The meaning of '**' indicates P value under 0.01. This explanation has been added in the caption of Table 2.

Reviewer 3 Report

Reviewer's report 

Date: 04 August, 2023 

Journal: International journal of molecular sciences 

Manuscript ID: ijms-2541129

Title: The complete mitochondrial genomes of two rock scallops (Bivalvia: Spondylidae): Insight into the gene rearrangements and adaptive evolution within Pectinoidea

Authors: Fengping Li, Yu Zhang, Tao Zhong, Xin Heng, Tiancheng Ao, Zhifeng Gu, Aimin Wang, Chunsheng Liu, Yi Yang *

The authors have sequenced two complete mitochondrial genomes of the rock scallops Spondylus versicolor and S. spinosus (Bivalvia: Spondylidae). Combined with mitochondrial genomes from GenBank, they used the Spondylus versicolor and S. spinosus genomes to reconstruct phylogenetic relationships within the superfamily Pectinoidea. 

Though there is nothing fundamentally incorrect with most of the analyses, I feel that at the present condition the paper is not appropriate for the International journal of molecular sciences. 

1). Abstract, lines 11-13: Consider English revision of the phrase “a cementing life habit”. 

For instance: “…the Spondylidae is a small group with one single genus that shares a sedentary (or sessile) habit of life cementing themselves to the substrate.” 

2). Pages 9-10: For the phylogenetic analysis of Pectinoidea the authors use only 14 mitochondrial genomes. However, GenBank contains around 70 complete mitochondrial genomes of Pectinoidea. It is not clear why the authors have limited the number of species in phylogenetic analysis. They note that “the phylogenetic of the Spondylidae to the other families in the Pectinoidea has been highly contentious.” It would be reasonable to analyze as many as possible species to clarify the phylogenetic relationships of Pectinoidea. 

3). Page 13: The authors present the branch-specific models supporting positive selection on the Spopndylidae lineage. The authors, however, did not use the site specific and branch-site specific models, which are important to explore differences in evolutionary patterns between separate genes in mitochondrial genomes studied. 

Abstract, lines 11-13: Consider English revision of the phrase “a cementing life habit”. 

For instance: “…the Spondylidae is a small group with one single genus that shares a sedentary (or sessile) habit of life cementing themselves to the substrate.” 

Author Response

Dear Reviewer,

First, we would like to say thank you for reviewing our manuscript and for your valuable comments. Our responses to the comments are shown below.

Comment 1: Abstract, lines 11-13: Consider English revision of the phrase “a cementing life habit”.

Response 1: Following the advice of the Reviewer, we have revised the sentence to “the Spondylidae is a small group with one single genus that shares a sedentary habit of life cementing themselves to the substrate”

Comment 2: Pages 9-10: For the phylogenetic analysis of Pectinoidea the authors use only 14 mitochondrial genomes. However, GenBank contains around 70 complete mitochondrial genomes of Pectinoidea. It is not clear why the authors have limited the number of species in phylogenetic analysis. They note that “the phylogenetic of the Spondylidae to the other families in the Pectinoidea has been highly contentious.” It would be reasonable to analyze as many as possible species to clarify the phylogenetic relationships of Pectinoidea.

Response 2: It is true that analyzing as many as possible species to clarify the phylogenetic relationships of Pectinoidea would be reasonable and we totally agree. Although GenBank contains around 70 complete mitochondrial genomes, some of them are the repeats of certain economic species in Pectinidae.

We noted that “the phylogenetic of the Spondylidae to the other families in the Pectinoidea has been highly contentious.” This is a consequent derived from previous phylogenetic studies. At this moment, only 16 mitogenomes restricted to two families (Pectinidae and Spondylidae) within Pectinoidea are available on GenBank.

Comment 3: Page 13: The authors present the branch-specific models supporting positive selection on the Spopndylidae lineage. The authors, however, did not use the site specific and branch-site specific models, which are important to explore differences in evolutionary patterns between separate genes in mitochondrial genomes studied.

Response 3: We agree with the Reviewer that exploring differences in evolutionary patterns between separate genes is very important. Actually we did perform the branch-site specific models, and we discovered plenty of sites under positive selection. However, we failed to reveal any patterns between individual genes, since all genes include positive selected sites. Therefore, we kept the branch-specific models, which is the most significant result supporting positive selection on the Spondylidae lineage.

Reviewer 4 Report

This is a good paper providing interesting taxonomic information, justifying publication.  I do not see specific weak points other than to move tables from introduction and over to the other chapters. Thus, Table 1 should be in the Materials and methods.

Author Response

Dear Reviewer,

First, we would like to say thank you for reviewing our manuscript, and also for your nice words. Our responses to your comments are shown below.

Comment 1: Table 1 should be in the Materials and methods.

Response 1: In theory Table 1 should be in Materials and methods parts. However, we have to mention some important species information in the introduction part. Therefore it was deposited in the first place where it was mentioned. If Table 1 was removed from Introduction part, it would go to the Results and discussion section since this part is in front of Materials and methods section, and Table 1 was also mentioned in Results and discussion section.

Round 2

Reviewer 2 Report

Dr Li and Collaborators have revised their manuscript and are resubmitting a slightly improved version.

In the section about directional selection please describe analyses that did not identify selection alongside analyses that did identify selection, and provide a plausible explanation for this observation.

The explanation at lines 83-89 is very vague and contradictory. Please rephrase.

Please report the remapping strategy in the methods section.

While I assume the coding part of the genome to be correctly assembled, I have some concerns over the long non coding regions, especially the one interested by the 'bump'.
Is there sufficient overlap between the Sanger sequences and the NGS assembled part of the genome? Is there is sufficient overlap between forward and reverse Sanger sequences of each fragment?
Was the genomic sequence searched for repeats or areas of simple sequence (e.g. ling microsatellites)? How? With which results? Was the non coding region, and the 'bump' region especially, blasted to see if they resamble some known sequence (biological or technical)? Is there any region along the genome where coverage drops below a 'safe' minimum (optimally 100x, minimum 10x) indicating a possible misassembly or non completeness of the sequence?

Figure S1 has to be redrawn. The coverage is scaled in a way that highlights the 'bumb' but does not visualize coverage in the rest of the genome well. Remember that readers may be interested in: a) evaluate the 'bump', but also b) check if coverage in the remaining part of the genome is sufficient and sufficiently stable to support a credible assembly. I suggest scaling up to show coverage along the genome and adding some graphical sign to identify the 'bump', reporting its coverage numerically on the image. Gene presence/absence and span is not consistent between figure S1 and table S2. I spotted a problem with Atp8 not showing in one genome, but please revise all.

Minor correction can be done at copyediting.

Author Response

Dear Reviewer,

Thanks again and we appreciate for the second-round comments provided for our manuscript. We have made corrections according to the advice. The responses are shown as following.

Point1: In the section about directional selection please describe analyses that did not identify selection alongside analyses that did identify selection, and provide a plausible explanation for this observation.

Response1: Actually we didn’t omit any analyses that did not identify selection. In this section, we performed branch-specific models suggested by the instructions of PAML software. And the branch-specific model has also been applicated in several studies (please refer to [17] and [19] in the manuscript). The differences between our study and previous studies [17, 19] is that we did not place the analyses using branch-site model in the manuscript. According to the instructions of PAML and previous studies, the branch-site model could be further used to identify the positions under positive selection when the analysis based on branch-specific models did not detect selection. We did not place the results of branch-site model because our analysis already revealed selection under branch-specific models.

We would like to describe the results of positive selection analysis based on branch-specific models in detail. The analysis using three different models (M0, M1 and M2) should be considered together. Firstly, the ω (Ka/Ks) ratio under the M0 (“one-ratio”) model is 0.03351, indicating that the strong functional constraints were evident in the 12 mitochondrial protein-coding genes of all the sampled pectinoids. However, the ω ratio averaged over all lineages is almost never > 1, since positive selection is unlikely to affect all sites over prolonged time. Thus, interest has been focused on detecting positive selection that affects only some lineages. The M1 (“free-ratios”) model fit the data significantly better than M0 model (p < 0.01), suggesting that the mitochondrial genes have been subject to different selection pressures in the different lineages of Pectinoidea. Further, the M2 (“two-ratios”) model was employed by setting family Spondylidae as a foreground branch. The Ka/Ks ratio (ω1=1.84124) of Spondylidae is significantly higher than other species (ω0=0.03263). The Ka/Ks ratio of Spondylidae is >1, therefore suggesting a positive selection under this branch.

The positive selection was found on the branch of Spondylidae against Pectinidae, indicating plenty of variations between the protein-coding genes of the two families. The Spondylidae and the Pectinidae possess quite different life habits. All members of the Spondylidae cement to substrates, while byssally attach is the major habit for the Pectinidae. The different life habits are related to different ecological requirements and behavioral attributes [45], which could lead to different mechanisms of energy metabolism. Therefore, the mitochondrial protein-coding genes which are related to oxygen usage and energy metabolism of animals, might have evolved toward different directions. On the other hand, since the sister relationship between the Spondylidae and the Pectinidae has been challenged in previous phylogenies [35], their phylogenetic positions within Pectinoidea may be different with the inclusion of more families. Since CodeML analysis is sensitive to tree topology, and future studies with a broader taxon sampling within Pectinoidea are needed to verify the present observed positive selection on the branch of Spondylidae within Pectinoidea.

Point2: The explanation at lines 83-89 is very vague and contradictory. Please rephrase.

Response2: We have rephrased the explanation as following.

“However, the bump in S. spinosus mitogenome also fell in the region generated by Sanger sequencing (Figure S1), which supported the absence of repetitive sequences. There is another possibility that only one copy of the repetitive sequences was amplified with PCR and sequenced using Sanger technology in the mitogenome of S. spinosus. The coverage bumps could be offset by adding different numbers of repeats according to the coverage of the reads, whereas such solution could lead to an excessive genome size which falls outside the normal range of Pectinoidea. Since the whole process of NGS assembly was visible and no errors were detected according to the mapped reads, it is assumed that the bumps (with a relatively higher AT content values) in the non-coding regions of two mitogenomes, might result from the flaw of NGS sequencing that regions with higher AT content values were tended to possess more sequencing depth.”.

Point3: Please report the remapping strategy in the methods section.

Response3: The remapping strategy is as following and added in the methods section.

“The short reads were remapped back to the assembled mitogenomes by Geneious Prime using the following parameters: custom sensitivity with a minimum mapping quality of 95%, a maximum mismatch of 5%, and fine-tuning 3 times.”

Point4: While I assume the coding part of the genome to be correctly assembled, I have some concerns over the long non coding regions, especially the one interested by the 'bump'.

Is there sufficient overlap between the Sanger sequences and the NGS assembled part of the genome? Is there is sufficient overlap between forward and reverse Sanger sequences of each fragment?

Was the genomic sequence searched for repeats or areas of simple sequence (e.g. ling microsatellites)? How? With which results? Was the non-coding region, and the 'bump' region especially, blasted to see if they resamble some known sequence (biological or technical)? Is there any region along the genome where coverage drops below a 'safe' minimum (optimally 100x, minimum 10x) indicating a possible misassembly or non completeness of the sequence?

Response4: When the Sanger Primers were designed, we have considered the overlap between Sanger and NGS parts. All the Sanger Primers were designed with around 100-150 overlapped bases. The Sanger fragment was 1,581 bp for S. versicolor and 415 bp for S. spinosus. For S. versicolor fragment, we sequenced the whole fragment in both directions. In each direction, the fragment was sequenced completely. The length of overlaps of S. versicolor is as almost the same length of this fragment. For S. spinosus fragment which is 415 bp in length, its overlap also covered all the fragment.  

We have searched the repeats of simple sequence using online tool Tandem Repeats Finder (https://tandem.bu.edu/trf/home), and we did detect some repetitive sequences but not related to the bumps. The non-coding regions have been blasted in GenBank, nevertheless, no results were found.

The bump in each mitogenome may be probably caused by the repetitive sequences. Even though the bump in S. spinosus mitogenome was sequenced by Sanger that did not support the discovery of repetitive sequences, it is also possible that PCR amplication only obtained one copy of these repetitive sequences. On the other hand, the bumps which were located in the non-coding regions all possess higher AT content, which might trigger the flaw of NGS sequencing that regions with higher AT content values were tended to possess more sequencing depth. We have discussed this part in the manuscript.

The sequencing depth has been added in the Figure S1. Even though there are some drops in both mitogenomes, their depth are all above 80.

Point5: Figure S1 has to be redrawn. The coverage is scaled in a way that highlights the 'bumb' but does not visualize coverage in the rest of the genome well. Remember that readers may be interested in: a) evaluate the 'bump', but also b) check if coverage in the remaining part of the genome is sufficient and sufficiently stable to support a credible assembly. I suggest scaling up to show coverage along the genome and adding some graphical sign to identify the 'bump', reporting its coverage numerically on the image. Gene presence/absence and span is not consistent between figure S1 and table S2. I spotted a problem with Atp8 not showing in one genome, but please revise all.

Response5: We have revised Figure S1. In the updated version, the sequencing depth was added across the whole mitogenome. The plot map of each mitogenome was divided in three parts, marked as fragment1, fragment2, and the bump between the two fragments. The sequencing depth (mean, maximum and minimum coverage) of all parts were indicated in the figure.

The Atp8 gene was missing in last version of figureS1, and now it has been added.

Reviewer 3 Report

Reviewer's report 

Date: 23 August, 2023 

Journal: IJMS

Manuscript ID: ijms-2541129 - Revised Version

Title: The complete mitochondrial genomes of two rock scallops (Bivalvia: Spondylidae): Insight into the gene rearrangements and adaptive evolution within Pectinoidea

Authors: Fengping Li, Yu Zhang, Tao Zhong, Xin Heng, Tiancheng Ao, Zhifeng Gu, Aimin Wang, Chunsheng Liu, Yi Yang *

The manuscript was significantly improved. It is appropriate now for the International journal of molecular sciences.

Author Response

Dear Reviewer,

Thanks for the second-round review, and we appreciate for your recommendation of our manuscript. And we have also made further revisions in the last version of the manuscript.

Best wishes,

Yi Yang